Alpha-centractin is a novel substrate of SETD3 methyltransferase in vitro

Witecka Apolonia 1
Emmel Paulina 1
Ślusarczyk Klaudia 1 2
Kamińska Julia Z. 1
Zaród Michał 1
Ishikawa Takao 3 4
Drożak Jakub j.drozak2@uw.edu.pl 1
1 Department of Metabolic Regulation, Faculty of Biology, Institute of Biochemistry, University of Warsaw , Warsaw , Poland
2 Doctoral School of Translational Medicine, The Centre of Postgraduate Medical Education , Warsaw , Poland
3 Department of Molecular Biology, Faculty of Biology, Institute of Biochemistry, University of Warsaw , Warsaw , Poland
4 Department of Environmental Microbiology and Biotechnology, Institute of Microbiology, Faculty of Biology, University of Warsaw , Warsaw , Poland
Gould Gwyn
Electronic publication date: 2025 Oct 20
Publication date: 2025
Volume: 13
Electronic Location ID: e20195
Received 2024 Jul 4; Accepted 2025 Sep 15
Copyright: ©2025 Witecka et al.
Copyright year: 2025
Copyright holder: Witecka et al.
License: This is an open access article distributed under the terms of the Creative Commons Attribution License, which permits unrestricted use, distribution, reproduction and adaptation in any medium and for any purpose provided that it is properly attributed. For attribution, the original author(s), title, publication source (PeerJ) and either DOI or URL of the article must be cited.
License URL: https://creativecommons.org/licenses/by/4.0/

Keywords: SETD3, Novel substrates, Human cell lines, Alpha-centractin, ARP1, Histidine methyltransferase

Funding: National Science Centre, the Republic of Poland 2017/27/B/NZ1/00161 Excellence Initiative BOB-661-1384/2024 European Union European Regional Development Fund This work was supported by the Opus-14 grant from the National Science Centre, the Republic of Poland, (2017/27/B/NZ1/00161), and by the “Excellence Initiative - Research University (IDUB)” Program from the Minister of Science and Higher Education, the Republic of Poland, (BOB-661-1384/2024). It was carried out using the CePT infrastructure financed by the European Union European Regional Development Fund within the Operational Program “Innovative Economy” for 2007–2013. There was no additional external funding received for this study. The funders had no role in study design, data collection and analysis, decision to publish, or preparation of the manuscript.

==============================
Background

The SETD3 enzyme, a protein histidine methyltransferase, catalyzes the Nτ-methylation of the histidine 73 residue in β-actin. This post-translational modification is important for maintaining cytoskeleton integrity, and actin remains the only known substrate of this methyltransferase to date. However, SETD3 was also postulated to play a role in the regulation of processes that are not directly related to actin homeostasis, such as cell cycle control and response to hypoxic conditions. These findings suggest that actin may not be the sole substrate of SETD3 methyltransferase. Here, we demonstrate that SETD3 methylates additional proteins in human cells, and α-centractin (ACTR1A) may be one of them.

Methods

Three different human SETD3 knockout cell lines (HAP1, HeLa, HEK293T) were generated with the CRISPR/Cas9 method and used as a source of SETD3 substrates. Fluorography was used to detect the SETD3-dependent methylation of proteins present in cell lysates, while the TurboID biotin ligase proximity labeling technique was used to isolate proteins that interact with SETD3. The molecular identity of the proteins was determined by mass spectrometry and the activity of recombinant SETD3 towards potential substrates was tested using a radiochemical assay.

Results

Fluorography revealed that SETD3 methylates at least five novel proteins besides β-actin in HAP1 cells. TurboID proximity labeling identified α-centractin, a key dynactin subunit, as an SETD3 interactor and an in vitro methylation target, suggesting that SETD3 potentially regulates not only actin cytoskeleton dynamics but also dynein-mediated intracellular transport.

Introduction

Protein histidine methylation is a post-translational modification (PTM) discovered in rabbit muscle actin in 1967 (Johnson, Harris & Perry, 1967). However, early evidence supporting its presence was reported at least a decade prior (Searle & Westall, 1951; Tallan, Stein & Moore, 1954). This PTM involves the attachment of a methyl group to the either Nπ (N1) or Nτ (N3) atom of the imidazole ring on a histidine residue within a protein (for a review, see Kwiatkowski & Drozak, 2020). Specific methyltransferases utilize S-adenosyl-L-methionine as a methyl donor to catalyze these reactions.

At physiological pH (around 7.0), both nitrogen atoms in the imidazole ring of L-histidine can be protonated, leading to two neutral tautomers (π and τ) existing in equilibrium (Kwiatkowski & Drozak, 2020). Consequently, histidine N-methylation effectively “locks” the molecule in one specific tautomeric form (either Nπ- or Nτ-methyl), with a fixed position for the proton-accepting nitrogen within the methylated imidazole ring (Kwiatkowski & Drozak, 2020). This seemingly simple PTM significantly alters the biochemical properties of the targeted histidine residue in the protein.

Although the presence of methylated histidine residues has long been documented for some proteins in nature, including mammalian β-actin (Nyman et al., 2002), S100A9 (Raftery et al., 1996), myosin (Elzinga & Collins, 1977), and myosin light chain kinase 2 (Meyer & Mayr, 1987), the biochemical and physiological significance of this PTM have remained largely unclear. This was at least in part due to a very limited knowledge on the biochemical and molecular properties of enzymes responsible for protein histidine methylation. Just a few years ago, SETD3 emerged as the first identified metazoan histidine methyltransferase. This actin-specific Nτ-methyltransferase regulates cytoskeletal assembly and smooth muscle contractility in mammals (Kwiatkowski et al., 2018; Wilkinson et al., 2019) (Fig. 1A).

Figure 1 The SETD3 enzyme is actin-specific histidine N-methyltransferase.

(A) SETD3 catalyzes Nτ-methylation of H73 residue in β-actin. S-adenosyl-L-methionine (SAM) serves as a methyl-group donor and is converted into S-adenosyl-L-homocysteine (SAH) following the transfer reaction. (B) The structure of human β-actin (PDB 2BTF) with a dissected peptide that spans T66 to H88 (cyan) and contains a methylated H73 residue (magenta). (C) The structure of SETD3 in complex with the methylated β-actin peptide (66–88, cyan) (PDB 6ICT). A close-up view of the β-actin peptide indicates that this fragment of β-actin adopts an extended conformation upon binding to SETD3.

These findings reignited interest in the field, leading to the discovery of several novel protein histidine methyltransferases. Notably, METTL9 was shown to methylate the Nπ position of the HxH motif in various mammalian proteins, including the immunomodulatory protein S100A9 and the NDUFB3 subunit of mitochondrial respiratory complex I (Davydova et al., 2021). METTL9-mediated methylation enhances complex I respiration, while Nπ-methyl histidine in S100A9 reduces its zinc-binding affinity (Davydova et al., 2021; Cao et al., 2024). Moreover, the human METTL18 enzyme was found to Nτ-methylate histidine 245 in ribosomal protein RPL3, ensuring proper translation elongation, and mirroring the function of its yeast homolog HPM1 (Webb et al., 2010; Al-Hadid et al., 2016; Małecki et al., 2021; Matsuura-Suzuki et al., 2022). Recently, mammalian carnosine N-methyltransferase 1 (CARNMT1) that catalyzes the conversion of histidine-containing dipeptide carnosine (β-alanyl-L-histidine) into anserine (β-alanyl-Nπ-methyl-L-histidine) in mammalian brain and muscle tissue (Drozak et al., 2015) was identified as a crucial Nπ-specific protein histidine methyltransferase, responsible for the modification of 20 different proteins in human cells (Shimazu et al., 2023). Notably, protein histidine methylation now appears ubiquitous in human cells, with hundreds of proteins undergoing methylation (Ning et al., 2016; Kapell & Jakobsson, 2021). This suggests the potential existence of unidentified protein histidine methyltransferases in the human proteome, or a broader substrate range for known enzymes than currently recognized.

The human SETD3 protein catalyzes Nτ-specific methylation of His73 in β-actin (Kwiatkowski et al., 2018; Wilkinson et al., 2019) (Fig. 1B). This modification is believed to facilitate actin polymerization. Consequently, loss of SETD3 activity leads to depletion of filamentous actin and disrupts cytoskeletal integrity, which appears particularly detrimental for proper uterine contraction in mammals (Kwiatkowski et al., 2018; Wilkinson et al., 2019).

SETD3 is a 600-amino acid protein containing SET and Rubisco LSMT substrate binding (RuBisCo-subs-bind) domains at its N- and C-terminus, respectively. The SET domain binds S-adenosyl-L-methionine (SAM), while the RuBisCo-subs-bind domain likely interacts with protein substrates. Since full-length β-actin could not be co-crystallized with SETD3, X-ray crystallography was used to determine the 3D structures of SETD3 bound to actin-derived His73 peptides (Guo et al., 2019; Dai et al., 2019) (Fig. 1C). These structures revealed that the His73-containing fragments of β-actin specifically interact with SETD3 in a narrow cleft formed by the SET domain, where SAM is also bound. Further structural analysis supported by biochemical experiments indicated that β-actin peptides interact with SETD3 through amino acid sequences flanking the His73 residue. Importantly, replacing any of these flanking residues with alanine significantly reduced SETD3 binding affinity. These findings suggest highly sequence-specific recognition and methylation of β-actin by SETD3, potentially making it the methyltransferase’s sole substrate (Guo et al., 2019; Dai et al., 2019).

However, it is important to note that protein methyltransferase activity and substrate specificity can often be influenced by interactions with specific non-substrate partner proteins (for review, see Herz, Garruss & Shilatifard, 2013). Therefore, some methyltransferases can modify different substrates depending on their interacting partners. Interestingly, Cohn et al. (2016) showed that human SETD3 interacts with approximately 170 different intracellular proteins, including actin. This suggests the potential for additional substrates for this enzyme in mammalian cells. However, this question remains largely unexplored.

In this study, we report the identification of α-centractin (ACTR1A) as a novel SETD3 substrate in vitro. ACTR1A is an actin-related protein and a subunit of the dynactin complex. We further present evidence suggesting the presence of additional SETD3 protein substrates beyond β-actin in human cells.

Materials & Methods

Materials

High-purity reagents (analytical grade whenever possible) were purchased from various suppliers: Sigma-Aldrich (USA), Roche Diagnostics (Switzerland), or Merck (Germany). Radiolabeled S-[methyl-3H]adenosyl-L-methionine ([3H]SAM) and scintillation cocktail Ultima Gold LSC were obtained from PerkinElmer (USA). The deuterated form, S-[methyl-2H]adenosyl-L-methionine ([2H]SAM), was purchased from C/D/N Isotopes Inc. (Canada). Synthetic peptides were procured from Caslo Laboratory (Denmark). HisTrap FF crude (pre-charged with Ni2+ ions) and PD-10 desalting columns were obtained from Cytiva (USA). Dowex 50W-X4 (200 mesh) ion-exchange resin was purchased from Sigma-Aldrich. Vivaspin-500 concentrator tubes were obtained from Sartorius Stedim (Germany). All other enzymes, DNA modifying enzymes, and the TurboFect transfection reagent were purchased from various suppliers: Thermo Fisher Scientific (formerly Thermo-Fermentas, USA), A&A Biotechnology (Poland), or BioShop (Canada).

SETD3 methyltransferase activity assay

Methylation of proteins by SETD3

The activity of SETD3 methyltransferase was measured using a previously described method (Kwiatkowski et al., 2018). Briefly, enzyme activity was determined by quantifying the incorporation of a [3H]-methyl group from S-[methyl-3H]adenosyl-L-methionine ([3H]SAM) into recombinant human β-actin or α-centractin substrates. The standard incubation mixture (110 µL) contained 25 mM Tris–HCl (pH 8.0), 10 mM KCl, 1 mM DTT, 2 µM protein substrate, 1.6 µg of purified recombinant S-adenosyl-L-homocysteine (SAH) nucleosidase (from E. coli; final concentration: 600 nM), and 1 µM [1H+3H]SAM (about 300 × 103 cpm). The reaction was initiated by adding the enzyme preparation and incubated at 37 °C for the time indicated in the figure legends.

Similar to assays for nonribosomal peptide synthetase activity (Richardt et al., 2003; Drozak et al., 2014), the incubation was terminated by adding 100 µL of the reaction mixture to a quenching solution containing 25 µL of bovine serum albumin (BSA) (1 mg/mL) and 800 µL of ice-cold 10% (w/v) trichloroacetic acid (TCA). After incubation on ice for 10 min, the precipitated proteins were collected by centrifugation (13,000× g, 10 min) and washed twice with ice-cold 10% TCA. The final pellet was then dissolved in 500 µl of pure formic acid. Solubilized pellets (450 µl) were mixed with 10 ml of scintillation fluid (Ultima Gold) and the incorporated radioactivity was analyzed with a Beckman LS6000 IC liquid scintillation counter.

Methylation of synthetic peptides by SETD3

The methylation of synthetic peptides as potential SETD3 substrates was assessed by measuring the incorporation of a [3H]-methyl group from [3H]SAM into the corresponding peptide. The incubation mixture composition mirrored that described above, except that protein substrates were replaced with 1 mM of each peptide. Control reactions lacking peptides were included in all assays.

The reaction was initiated by adding the enzyme preparation and incubated at 37 °C for the time indicated in the figure legends. The incubation was terminated by adding 100 µL of the reaction mixture to 200 µL of ice-cold 10% (w/v) perchloric acid (HClO4). Samples were then diluted with 120 µL of water and centrifuged at 13,000× g for 10 min. Following neutralization of the supernatant with a 1:1 mixture of 3 M K2CO3 and 3 M KOH, the salts were removed by centrifugation (13,000× g for 10 min). The clarified supernatant was diluted 5-fold with 30 mM sodium formate buffer (pH 3.5) and 2 mL aliquots were loaded onto Dowex 50W-X4 columns (1 mL bed volume, pre-equilibrated with 30 mM sodium formate buffer, pH 3.5).

The columns were washed with 3 × 2 mL aliquots of the equilibration buffer. The eluate containing methylated peptide IARS1 was collected. Subsequently, methylated peptides ACTB and ACTR1A, along with any remaining non-methylated peptides, were eluted with 4 × 2 mL aliquots of 20 mM HEPES buffer (pH 7.5) containing 0.5 M NaCl. Finally, to remove any unconsumed [3H]SAM, the columns were washed with 3 × 2 mL aliquots of 1 M ammonium hydroxide (NH4OH).

For scintillation counting, all samples were mixed with 10 mL of scintillation cocktail (Ultima Gold), and incorporated radioactivity was quantified using a Beckman LS6000 IC liquid scintillation counter.

Protein purification

Expression and purification of recombinant SETD3 proteins

Human wild-type SETD3 plasmid (pCOLD I-SETD3) was prepared as previously described (Kwiatkowski et al., 2018). A mutated form of SETD3 enzyme (R253A) was generated using a QuikChange II XL kit (Agilent) with pCOLD I-SETD3 as a template and mutagenic primers listed in Table 1. The construct encoding the R253A variant was verified by DNA sequencing (Eurofins, Germany). The R253A mutant was then produced in E. coli using the same procedure established for wild-type SETD3.

Table 1 Sequences of primers used in PCR experiments.

Primer	Sequence	Restriction site	Plasmid	Protein(s) expressed	
CRISPR/Cas 9 construct preparation	
CRISPR-hSETD3-S1	CACCGgttctgtcgagggttttgaaa	BbsI	pSpCas9n(BB)-2A-Puro (PX462) V2.0	not applicable	
CRISPR-hSETD3-AS1	AAACtttcaaaaccctcgacagaacC	BbsI	pSpCas9n(BB)-2A-Puro (PX462) V2.0	not applicable	
CRISPR-hSETD3-S2	CACCGgaggcccatttcattagatc	BbsI	pSpCas9n(BB)-2A-Puro (PX462) V2.0	not applicable	
CRISPR-hSETD3-AS2	AAACgatctaatgaaatgggcctcC	BbsI	pSpCas9n(BB)-2A-Puro (PX462) V2.0	not applicable	
TurboID amplification	
C-TurboID-S	gagtctagaATGAAAGACAATACTGTGCCTC	XbaI	pEF6Myc-HisA/SETD3	C-terminal TurboID and SETD3	
C-TurboID-AS	ttgttcgaaCTCGAGTGCGGCCGCAAG	BstBI	pEF6Myc-HisA/SETD3	C-terminal TurboID and SETD3	
N-TurboID-S	taaggtaccgccaccATGAAAGACAATACTGTGCCTC	KpnI	pEF6Myc-HisA/SETD3	N-terminal TurboID and SETD3	
N-TurboID-AS	ggcggtaccCTCGAGTGCGGCCGCAAG	KpnI	pEF6Myc-HisA/SETD3	N-terminal TurboID and SETD3	
α-Centractin (ACTR1A) amplification	
ACTR1A-S	tatacatATGGAGTCCTACGATGTGATC	NdeI	pCOLD I	N-terminal His6-tagged α-centractin	
ACTR1A-AS	tatactcgagTTAGAAGGTTTTTCTGTGGATG	XhoI	pCOLD I	
Site-directed mutagenesis of the SETD3 enzyme	
R253A-SETD3-S	CTCTTCTGTTATGACGGCGCAAAACCAAATTCCCAC	N/A	pCOLD I	N-terminal His6-tagged mutated form of SETD3	
R253A-SETD3-AS	GTGGGAATTTGGTTTTGCGCCGTCATAACAGAAGAG	N/A	pCOLD I	N-terminal His6-tagged mutated form of SETD3	
Notes.

The nucleotides corresponding to the coding sequences are in capital letters, the Kozak consensus sequence is shown in bold, the added restriction sites are underlined, and mutated codons are shown in bold and italic.

Recombinant His6-tagged human SETD3 (wild-type and R253A) were produced and purified with slight modifications to the method by Kwiatkowski et al. (2018). Briefly, proteins were overexpressed in E. coli BL21(DE3) cells overnight at 16 °C and 200 rpm following induction with 0.3 mM IPTG. Purification involved a HisTrap FF crude column (5 mL) with a stepwise imidazole gradient (15 mL of 60 mM, 16 mL of 150 mM, and 20 mL of 300 mM) in a buffer containing of 50 mM HEPES (pH 7.5), 400 mM NaCl, 10 mM KCl, and 1 mM DTT.

While both proteins were present in the 150 mM and 300 mM imidazole fractions (see Fig. S1), only the highly purified (>95% purity by sodium dodecyl sulfate polyacrylamide gel electrophoresis) fraction eluted at the highest imidazole concentration was used for further experiments. The elution buffer was exchanged by sequential dialysis against 500 mL of dialysis buffer (20 mM Tris–HCl pH 7.5, 200 mM NaCl, 1 mM DTT, and 6% sucrose) overnight at 4 °C, followed by two additional 3-hour dialyzes against 500 mL of the same buffer at room temperature. The yield of purified recombinant enzymes ranged from 3.4 to 6.2 mg of protein per 950 mL of culture. Finally, the purified proteins were aliquoted and stored at −80 °C (Al-Fakhar et al., 2023).

Expression and purification of recombinant β-actin from inclusion bodies

Overexpression and purification of β-actin from E. coli inclusion bodies was performed precisely as described by Kwiatkowski et al. (2018) and Drozak (2020).

Expression and purification of recombinant α-centractin from inclusion bodies

The coding sequence (Open Reading Frame, ORF) for human α-centractin (NM_005736.4) was obtained from the DNASU Plasmid Repository (clone ID: HsCD00730956). Pfu DNA polymerase and specific primers flanking the ORF were used for polymerase chain reaction (PCR) amplification. The primers included the start and stop codons and restriction enzyme sites for cloning (listed in Table 1). The amplified fragment was then cloned into the pCOLD I vector (Takara Bio) for expression of the protein with an N-terminal His6 tag in E. coli. The construct was verified by DNA sequencing (Eurofins).

For α-centractin production, E. coli BL21(DE3) cells (Agilent, USA) were transformed with the plasmid and a single colony was used to inoculate an overnight pre-culture. Subsequently, 10 mL of the pre-culture was used to inoculate 1 L of LB broth containing 100 µg/mL ampicillin. The culture was incubated at 37 °C and 200 rpm until the optical density at 600 nm (OD600) reached 0.6. The culture was then chilled on ice for 20 min to induce a cold shock, followed by the addition of IPTG to a final concentration of 0.25 mM to induce protein expression. Cells were incubated for an additional 20 h at 17 °C and 200 rpm before harvesting by centrifugation (6,000 × g for 10 min).

The cell pellet was resuspended in 50 mL of lysis buffer containing 20 mM HEPES (pH 7.5), 1 mM DTT, 1 mM ADP, 0.5 mM PMSF, 2 µg/mL leupeptin, 2 µg/mL antipain, 0.2 mg/mL lysozyme, and 1,000 U Viscolase. Cell lysis was achieved by freezing in liquid nitrogen, followed by thawing and vortexing (three times). The lysate was then clarified by centrifugation at 20,000 × g for 30 min at 4 °C.

The resulting pellet, enriched for inclusion bodies, was resuspended in 50 mL of buffer A (20 mM HEPES, pH 7.5, 2 M urea, 0.5 M NaCl, 5 mM DTT, 2 mM EDTA) using a Potter-Elvehjem homogenizer. After centrifugation at 20,000 × g for 10 min at 4 °C, the pellet was washed sequentially with one 50 mL aliquot of buffer B (20 mM HEPES, pH 7.5, 0.5 M NaCl, 5 mM DTT, 2 mM EDTA) and one 50 mL aliquot of buffer C (20 mM HEPES, pH 7.5, 0.5 M NaCl).

The washed inclusion bodies were then solubilized in 50 mL of loading buffer (20 mM Tris–HCl, pH 7.5, 6 M guanidine HCl, 0.5 M NaCl, 30 mM imidazole) and loaded onto a HisTrap FF crude column (5 mL) pre-equilibrated with the same buffer.

The column was washed with 30 mL of loading buffer. Bound protein was refolded by applying a linear gradient of decreasing guanidine HCl concentration in loading buffer (from 6 M to 0 M over 20 min). The column was then washed with 15 mL of loading buffer without guanidine HCl. Finally, the retained proteins were eluted with a stepwise imidazole gradient (40 mL of 60 mM and 18 mL of 300 mM). Fractions containing the recombinant α-centractin, as confirmed by SDS-PAGE (Fig. S2), were eluted in homogeneous form with 300 mM imidazole. The purified protein was immediately dialyzed against a buffer containing 20 mM Tris–HCl, pH 7.5, 1 mM DTT, 6% sucrose, 2 µg/mL leupeptin, and 2 µg/mL antipain. The yield of recombinant α-centractin ranged from one to three mg per liter of culture. The purified protein was stored at −80 °C.

Generation of SETD3 knockout cell lines using CRISPR/Cas9

Cell lines and culture conditions

HAP1 cells (Horizon Discovery, UK).

These near-haploid cells have a disomy of chromosome 8 (25, XY, +8, Ph+) and express SETD3 at 37.5 nTPM (Human Protein Atlas). We used previously described HAP1 SETD3 knockout (KO) clones (Kwiatkowski et al., 2018).

HEK293T and HeLa cells (Sigma-Aldrich).

These hypotriploid cell lines have modal chromosome numbers of 64 and 76–80 for HEK293T and HeLa, respectively. They express SETD3 at 20.6 nTPM and 34.2 nTPM, respectively (Human Protein Atlas). We confirmed SETD3 expression in our cultured cells (Fig. S3).

HAP1 cells were cultured in IMDM supplemented with 10% FBS, two mM L-glutamine, and penicillin/streptomycin (Thermo Fisher Scientific). HeLa cells were cultured in EMEM with 10% FBS, two mM L-glutamine, and penicillin/streptomycin. HEK293T cells were cultured in DMEM with penicillin, streptomycin, and 10% FBS. All cells were maintained at 37 °C in a humidified incubator with 95% air and 5% CO2.

CRISPR/Cas9 construct design and transfection

CRISPR/Cas9 constructs targeting exon 4 of the human SETD3 gene were generated. Two different sets of annealed primer pairs (sequences in Table 1) were ligated into the BbsI-digested pSpCas9n(BB)-2A-Puro (PX462) V2.0 vector (Addgene plasmid #62987, a gift from F. Zhang, Massachusetts Institute of Technology).

Cells were transfected with the CRISPR constructs using Fugene HD (Promega, Germany) transfection reagent as previously described (Collard et al., 2016). Puromycin selection (2–4 µg/mL) was initiated 24 h post-transfection and maintained for 11 days with media changes every other day.

Isolation and genotyping of SETD3 knockout clones

Single puromycin-resistant colonies were isolated and expanded. After trypsinization, cell suspensions were seeded in 96-well plates. Once confluent, cells were passaged and used for either genotypic analysis or frozen for future use.

Genomic DNA was isolated from puromycin-resistant clones using the Extractme Genomic DNA kit (Blirt). The targeted regions were amplified by PCR, and the products were sequenced to identify mutations. Three clones from each cell line with frameshift mutations leading to premature stop codons were selected for further studies. Mutation analysis was performed using the DSDecode tool (Liu et al., 2015) for automated decoding of superimposed chromatograms and verified using the Indigo tool from the GEAR website (Rausch et al., 2020) (https://www.gear-genomics.com).

Western blot confirmation of SETD3 knockout

The absence of SETD3 protein in each KO cell line was also confirmed by western blot analysis (Fig. S3). The protocol followed previously described methods (Drozak et al., 2013) using a rabbit polyclonal primary antibody against human SETD3 (ab174662, Abcam, UK; 1:4,000 dilution) and a horseradish peroxidase-conjugated goat anti-rabbit secondary IgG antibody (AS09602, Agrisera, Sweden; 1:20,000 dilution).

Fluorographic detection of SETD3 activity

In Vitro methylation reaction

SETD3 wild-type (WT) or knockout (KO) cells were lysed in buffer containing 50 mM Tris–HCl (pH 7.4), 80 mM NaCl, 20 mM KCl, 1 mM MgCl2, 1 mM DTT, 0.01% Tween 20, 2 µg/mL leupeptin, 2 µg/mL antipain, and 30 mM imidazole. The lysate was subjected to three freeze-thaw cycles in liquid nitrogen (LN2) followed by sonication (five 1-minute pulses with 30-second breaks on ice) and centrifugation (10 min, 12,000 × g, 10 °C). Finally, 30 units/mL of Viscolase was added to the supernatant.

Cell lysates (300 µg protein) were pre-incubated for 30 min at 37 °C with a reaction mixture containing 10 µM MnCl2, 0.6 µM S-adenosylhomocysteine (SAH) hydrolase, 0.6 µM adenine deaminase, and 5 µM [1H]SAM to allow endogenous protein methyltransferases to modify their substrates. The pre-incubated lysates were then purified using Zeba Spin Desalting Columns (Thermo Scientific) according to the manufacturer’s instructions. MnCl2 (10 µM) was added to the desalted eluate, followed by human recombinant SETD3 protein (10 µg), and either R253A protein (10 µg) or bovine serum albumin (BSA) as the controls. The reaction was initiated by adding either 1 × 106 cpm [3H]SAM or 200 µM deuterated SAM ([2H]SAM). The incubation proceeded for 60, 120 or 180 min at 37 °C.

After incubation, the reactions were stopped by adding 4x concentrated Laemmli sample buffer and heating at 45 °C for 10 min. Samples (approximately 70 µg protein) were resolved by SDS-PAGE and stained with Coomassie Brilliant Blue.

Fluorography and protein identification

Gels were then processed for fluorography. Briefly, gels were soaked in Amplify fluorographic reagent (Amersham, GE Healthcare, UK) for 25 min at room temperature with shaking in the dark. Following soaking, gels were dried on Whatman blotting paper under vacuum at 60 °C (650 mbar) and exposed to Hyperfilm ECL (GE Healthcare, USA) in a cassette (Kodak, USA) for 30 days at −80 °C. The exposed film was then developed for visualization.

Protein bands containing deuterated proteins that corresponded to the [3H]-labeled bands on the fluorograms were excised from the polyacrylamide gels and digested with trypsin. Details on in-gel digestion of peptides and subsequent protein identification by tandem mass spectrometry are provided in the “Identification of proteins by tandem mass spectrometry” section.

TurboID-based proximity labeling of proteins

DNA constructs

The coding sequence for the TurboID biotin ligase was PCR-amplified from the TurboID-His6_pET21a plasmid (Addgene #117177, Table 1). The amplified fragment was then subcloned into the pEF6Myc-HisA/SETD3 plasmid (Kwiatkowski et al., 2018) using restriction enzyme sites KpnI or BstBI and XbaI to generate constructs encoding fusion proteins with TurboID ligase at either the N-terminus or C-terminus of SETD3.

Biotinylation and pulldown

HAP1 SETD3 KO, HEK293T SETD3 KO, and HeLa SETD3 KO cells were cultured as described above. Cells were co-transfected with both TurboID-SETD3 constructs (N- and C-terminal) using TurboFect transfection reagent (Thermo Fisher) according to the manufacturer’s protocol. After 48 h, cells were incubated with 50 µM biotin for 30 min. Cells were then washed with PBS and lysed in 350 µL of lysis buffer (50 mM Tris–HCl, pH 7.4, 80 mM NaCl, 20 mM KCl, 1 mM MgCl2, 1 mM DTT, 2 µg/mL leupeptin/antipain, and 0.01% Tween 20) by freezing in liquid nitrogen, followed by thawing and vortexing (three times).

Biotinylated proteins were enriched using a modified BioID pulldown procedure (Kim et al., 2016). Briefly, one mL of cell lysate was incubated with 50 µL of Pierce magnetic beads (Thermo Scientific) overnight at 4 °C with constant mixing. The beads were collected using a magnetic stand and washed sequentially with the following buffers: washing buffer (50 mM Tris–HCl, pH 7.4; 40 mM NaCl; 20 mM KCl; 1 mM MgCl2; 1 mM DTT; 2 µg/mL leupeptin/antipain), high-salt buffer (0.2 M KCl, 0.1 M Na2CO3, 2 M urea in Tris–HCl, pH 7.4), and washing buffer again (twice).

Analysis of biotinylated proteins

Half of the enriched sample was resuspended in a Laemmli sample buffer, heated at 95 °C for 5 min, and the supernatant collected for western blot analysis. The other half was digested with Trypsin Gold (Promega, Germany) for protein identification by nano-Ultra-Performance Liquid Chromatography coupled with Quadrupole Time-of-Flight mass spectrometry (UPLC-Q-TOF) analysis. Details on in-gel digestion of peptides and subsequent protein identification by tandem mass spectrometry are provided in the “Identification of proteins by tandem mass spectrometry” section.

The presence of biotinylated proteins was confirmed by western blot analysis as previously described (Drozak et al., 2013) using a rabbit anti-biotin monoclonal primary antibody (D5A7, Cell Signaling, 1:1,000 dilution) and a horseradish peroxidase-conjugated goat anti-rabbit secondary IgG antibody (AS09602, Agrisera, Sweden; 1:10,000 dilution).

Protein labeling for gel filtration chromatography

In vitro methylation

Eleven culture plates (10 cm diameter) containing HAP1 SETD3 KO cells were harvested. Cells were scraped and resuspended in a total of 11 mL lysis buffer (50 mM Tris–HCl, pH 7.4, 40 mM NaCl, 20 mM KCl, 1 mM MgCl2, 1 mM DTT, 0.01% Tween 20, 2 µg/mL leupeptin, and 2 µg/mL antipain). The cell suspension was lysed by three freeze-thaw cycles in liquid nitrogen (LN2). The lysate was then sonicated (five 1-minute pulses with 30-second breaks on ice) and centrifuged (10 min, 12,000 × g, 10 °C). Finally, 30 units/mL of Viscolase was added to the supernatant.

Cell lysate (12 mg protein) was supplemented with 10 µM MnCl2, 0.6 µM S-adenosylhomocysteine (SAH) hydrolase, 0.6 µM adenine deaminase, and 10 µM [1H]SAM. The mixture was pre-incubated for 40 min at 37 °C to allow endogenous protein methyltransferases to perform their methylation activity. After pre-incubation, the methylation buffer was exchanged to a new buffer (50 mM Tris–HCl, pH 8.0, 20 mM KCl, 1 mM MgCl2, 1 mM DTT, 10 µM MnCl2, 1 µg/mL leupeptin, and 1 µg/mL antipain) using PD-10 desalting columns (Cytiva) according to the manufacturer’s instructions.

The protein labeling reaction was initiated by adding either 400 µg of human recombinant SETD3 protein (experimental sample) or BSA (control sample) to 2.5 mL of cell lysate, along with 7 × 106 cpm [3H]SAM or 40 µM [2H]SAM. The reactions proceeded for 180 min at 37 °C. Following incubation, the reaction mixtures were desalted using PD-10 desalting columns (Cytiva) to remove unreacted [3H]SAM. The desalted eluates were concentrated using Vivaspin-500 ultrafiltration devices (Millipore Sigma) to a final volume of approximately 1.1 mL (three mg protein, 0.73 × 106 cpm) for [3H]-labeled control lysate, 0.7 mL for [3H]-labeled lysate (SETD3), and 0.7 mL for [2H]-labeled lysate (SETD3). The [2H]- and [3H]-labeled lysates (SETD3) were then pooled, resulting in a final volume of 1.4 mL (eight mg protein, 1.85 × 106 cpm).

Gel filtration chromatography and analysis of radioactive fractions

One milliliter of the pooled, [2H + 3H]-labeled lysate was loaded onto a Superdex 200 16/60 column (GE Healthcare) equilibrated with buffer containing 50 mM Tris–HCl (pH 7.5), 150 mM NaCl, 10 mM KCl, 1 mM DTT, and 1 mM MgCl2. The column was eluted with 148 mL of equilibration buffer, collecting 1.5 mL fractions. Five hundred microliters of each fraction were mixed with 10 mL of scintillation fluid (Ultima Gold, PerkinElmer), and the incorporated radioactivity was measured using a Beckman LS6000 IC liquid scintillation counter.

Fractions exhibiting the highest radioactivity (F14-F17) were further concentrated 10 or 40 times using Vivaspin-500 ultrafiltration devices. Ten microliters of the concentrated samples were mixed with 4x Laemmli sample buffer, separated by SDS-PAGE, and analyzed by fluorography (a film exposed for two weeks at −80 °C). Proteins present in fraction 15 were subjected to tryptic digestion, followed by identification using Liquid Chromatography with tandem mass spectrometry (LC-MS/MS) as described in the “Identification of proteins by tandem mass spectrometry” section.

Identification of proteins by tandem mass spectrometry

Protein bands visualized on fluorograms (10% polyacrylamide gel) were excised and subjected to in-gel digestion with Trypsin Gold (Promega, Germany) according to the method described by Shevchenko et al. (2006).

For protein identification from fractions obtained in TurboID pulldown experiments and gel filtration of radiolabeled cell lysates (Superdex 200), in-solution digestion was performed. Briefly, 17–34 µg of protein from fraction 15 or streptavidin magnetic beads were reduced with 5 mM DTT in a reaction mixture containing 100 mM ammonium bicarbonate (NH4HCO3) and 0.1% RapiGest (Waters, USA) at 50 °C for 30 min. The reaction mixture was then alkylated with 9 mM iodoacetamide for 40 min at room temperature. Subsequently, 2 µg of Trypsin Gold was added, and the digestion proceeded overnight at 30 °C with continuous shaking for on-bead digestion.

Peptides resulting from tryptic digestion were analyzed by nano-UPLC tandem mass spectrometry using an Acquity nano-UPLC system coupled with a Synapt G2 HDMS Q-TOF mass spectrometer (Waters, Milford, USA) equipped with a nanospray source. The instrument was operated in MSE; mode under default parameters. Briefly, 1−2.5 µg of digested peptides were loaded onto a Waters Symmetry C18 trapping column (20 mm × 180 µm) connected to a Waters BEH130 C18 UPLC column (250 mm × 75 µm). Peptides were eluted with a gradient of 1–85% acetonitrile in water (both containing 0.1% formic acid) at a flow rate of 0.3 µL/min. Eluted peptides were directly introduced into the mass spectrometer.

Data acquisition and analysis were performed using MassLynx 4.1 software (Waters, USA) and ProteinLynx Global Server 2.4 software (PLGS, Waters, USA) with a False Discovery Rate (FDR) set to ≤ 4%. Protein identification was achieved by searching a database containing the complete human (Homo sapiens) reference proteome downloaded from the NCBI Protein database, which was then randomized for the MS/MS software.

Detection of in vitro methylated α-centractin by tandem mass spectrometry

To obtain a sufficient quantity of methylated α-centractin, formed through the reaction catalyzed by recombinant human SETD3 protein, for subsequent mass spectrometry analysis, the reaction mixture was scaled up. In brief, 20 µg of recombinant human α-centractin was incubated for 2 h at 37 °C in 30 µL of a reaction mixture. This mixture contained 25 mM Tris–HCl, pH 8.0, 10 mM KCl, one mM DTT, 20 µM MnCl2, 30 µM [2H]SAM, 1.5 µg SAH nucleosidase, and either 1.5 µg SETD3 or 1.5 µg R253A protein. The incubation was terminated by snap-freezing in liquid nitrogen, and the reaction mixtures were stored at −70 °C.

For MS/MS analysis of deuterated protein, 19 µl of the flash-frozen sample (12.6 µg protein) was trypsin-digested ‘in solution’ for 16 h at 30 °C in 30 µl of the reaction mixture containing 100 mM NH4HCO3, 0.05% RapiGest (Waters), 1.7 mM TCEP, and 2 µg trypsin (MS grade, Promega). The digestion reaction was stopped by adding trifluoroacetic acid to a final concentration of 1%. The resulting peptides were then analyzed by nano-UPLC-tandem mass spectrometry, following the procedure detailed in the “Identification of proteins by tandem mass spectrometry” section. Detection of the trideuterium-methylated peptides was carried out in fully automated mode using PLGS 2.4 software (Waters). This software had been updated to include trideuterium-methylation (+17.03448 Da) of Cys, Asp, Asn, His, Lys, Arg, Glu, and Gln residues as a potential peptide modification.

Analytical methods

Sodium dodecyl sulfate polyacrylamide gel electrophoresis (SDS-PAGE) was performed using 10% polyacrylamide gels and a Tris-glycine running buffer. Following electrophoresis, the gels were stained with Coomassie Brilliant Blue Protein Staining Solution (Thermo) to visualize proteins. Protein concentration was determined spectrophotometrically using the Bradford assay (1976). Bovine serum albumin (BSA) served as a standard for the assay.

Results

SETD3 methylates targets other than β-actin

Previous autoradiography experiments using total cytoplasmic extracts from HT1080 cells expressing endogenous SETD3 revealed β-actin as the sole methylated protein (Wilkinson et al., 2019). However, this approach might have only detected highly abundant substrates.

To identify potential minor substrates for SETD3, we performed in vitro methylation assays using cell-free lysates from human HAP1 cells lacking SETD3 (SETD3 KO). Lysates were first preincubated with [methyl-1H]SAM to allow endogenous methyltransferases to modify their substrates. Subsequently, [methyl-3H]SAM was added along with recombinant SETD3 or BSA (control), and protein radiolabeling was performed. Fluorography of proteins resolved by SDS-PAGE revealed several radiolabeled bands with molecular weights of approximately 240, 160, 120, 43, 34, and 29 kDa, present predominantly in lysates containing recombinant SETD3 (Fig. 2). The prominent 43 kDa band likely corresponded to β-actin, while the remaining bands potentially represented novel, less abundant SETD3 substrates. To further substantiate these findings and exclude potential contamination of the radiolabeled bands by proteins from the SETD3 preparation, we conducted a modified fluorographic experiment. In this iteration, BSA was substituted with R253A SETD3, a catalytically inactive mutant of SETD3. Concurrently, radiolabeling of cell-free lysates from wild-type HAP1 cells with SETD3 was performed to compare the degree of protein methylation between SETD3 wild-type and knockout cells. Subsequent fluorography of SDS-PAGE-resolved proteins confirmed the presence of several radiolabeled bands specifically methylated by SETD3, albeit with signals that were more diffuse and less pronounced compared to the initial experiments (Fig. S4). In conclusion, these results suggested the presence of additional SETD3 substrates beyond β-actin in human cells.

Figure 2 SETD3 methylates proteins different from β-actin in human cells.

Cell-free lysates of human HAP1 cells (300 μg protein) were preincubated with [1H]SAM for 30 min and then gel-filtered to remove [1H]SAM. Next, [3H]SAM (1  ×  106 cpm) was added to the reaction mixture and the radiolabeling of the protein was started by adding homogeneous recombinant SETD3 or BSA (Ctrl) (10 μg protein). The reaction mixtures were incubated for 60 or 120 min at 37 °C and resolved by SDS-PAGE (A). The [3H]-methylated proteins were detected by fluorography for 4 weeks at −80 °C as described in Materials and Methods (B). The arrows show the methylated polypeptides corresponding to molecular weights of about 240 (Band 1), 160 (Band 2), 120 (Band 3), 43 (Band 4), 34 (Band 5), and 29 kDa (Band 6). The indicated bands were cut out from the gel obtained in a parallel experiment, in which [3H]SAM was replaced by [2H]SAM, and analyzed by mass spectrometry. M, prestained protein marker.

To identify proteins modified by SETD3, deuterated SAM ([methyl-2H]) was used instead of tritiated SAM as the methyl donor to distinguish SETD3-catalyzed methylation from other sources. In vitro labelled bands were cut out of the SDS-PAGE gel and analyzed by tandem mass spectrometry (Q-TOF). This analysis identified numerous proteins listed in File S1. Notably, it included condensin complex subunit 1 (NP_055680.3, Band 2), isoleucine tRNA ligase (NP_001361230.1, Band 2), and actin (NP_001186883.1, Band 4), all with specific histidine residues methylated.

However, mass spectrometry analysis only successfully identified labeled proteins in two of the six analyzed bands. This was likely due to the limited amount of protein present in the excised bands. To overcome this limitation and improve protein yield for mass spectrometry, we fractionated cell-free lysates labeled with both [methyl-2H] and [methyl-3H]SAM on a Superdex-200 gel filtration column. This aimed to partially purify SETD3 substrates before mass spectrometry analysis. Surprisingly, nearly all radiolabeled proteins eluted in the void volume of the column. This suggests that these proteins might exist within the lysates as protein complexes exceeding 600 kDa in molecular mass (Figs. 3A and 3B). The labeled fractions were then resolved by SDS-PAGE, and the methylated polypeptides were visualized using fluorography. Notably, the 43 kDa band corresponding to actin displayed the strongest signal, indicating it as the most prominently methylated protein (Fig. 3C). The remaining SETD3 substrates were only clearly detectable in the most radioactive fraction (fraction 15) after a 40-fold concentration step (Fig. 3D). Mass spectrometry analysis of fraction 15 revealed the presence of numerous proteins, including SETD3, actin, and several deuterated polypeptides (File S2). Interestingly, the presence of SETD3, with a native molecular weight of approximately 130 kDa, in fraction 15 suggests its potential involvement in the high-molecular-weight protein complexes eluting in the void volume of the Superdex 200 column.

Figure 3 SETD3 methylated proteins form complexes of high molecular weight.

(A) The protein standard mixture containing bovine thyroglobulin (670 kDa), human IgA (300 kDa), human IgA (150 kDa), ovalbumin (44 kDa) myoglobin (17 kDa) and uridine (0.24 kDa) was gel filtered on the Superdex 200 column for calibration. (B) Cell-free lysates of human HAP1 cells (2.5 mg protein) were preincubated with 10 μM [1H]SAM for 40 min, and then gel-filtered to remove [1H]SAM. Next, [3H]SAM (7 × 106 cpm) or 40 μM [2H]SAM were added to the reaction mixture and protein labeling started by adding homogenous recombinant SETD3 or BSA (Ctrl) (400 μg protein). The reaction mixtures were incubated for 180 min at 37 °C and desalted by gel filtration to remove [3H]SAM. Protein preparations labeled in the presence of SETD3 were pooled (2H+3H SETD3) and fractionated on the Superdex 200 column. In addition, the control preparation of proteins incubated in the presence of BSA (3H Ctrl), instead of the enzyme, was fractionated in the column. The indicated fractions were concentrated 10- or 40-fold by ultrafiltration (Vivaspin 500), and the [3H]-methylated proteins were detected by fluorography for 2 weeks at −80 °C as described in Materials and Methods (C, D). The arrows show the methylated polypeptides corresponding to molecular weights of about 240 (Band 1), 160 (Band 2), 120 (Band 3), 43 (Band 4), 40 (Band 5), and 29 kDa (Band 6).

SETD3 interacts with several proteins in human cells

Protein methyltransferase activity and substrate preference can be influenced by interactions with non-substrate partner proteins. Some methyltransferases exhibit context-dependent substrate methylation based on their interacting partners (Herz, Garruss & Shilatifard, 2013). Therefore, we investigated potential in vivo interactions between SETD3 and other proteins using a TurboID proximity labeling method (Branon et al., 2018). This technique employs a TurboID biotin ligase that rapidly conjugates biotin to nearby proteins. Subsequently, these biotinylated proteins can be isolated and identified by mass spectrometry (MS/MS).

We constructed two plasmids encoding SETD3 fused to the TurboID ligase at either the N or C-terminus. HEK293T cells lacking endogenous SETD3 (SETD3 KO) were transfected with both constructs for 48 h. Following a 30-minute TurboID labeling step, cells were lysed, and biotinylated proteins were enriched using streptavidin beads. Half of the beads were used for protein elution with biotin, and the eluted proteins were analyzed by Coomassie blue staining (Fig. 4A) and western blotting with an anti-biotin antibody (Fig. 4B). The remaining beads underwent tryptic digestion to generate peptides for LC-MS/MS analysis for protein identification.

Figure 4 SETD3 interacts with several proteins in human cells.

HEK293T cells were transfected for 48 h with an empty vector (Control) or vectors encoding the SETD3 enzyme in C- or N-terminal fusion with TurboID biotin ligase (TurboID). At 48 h after transfection, cells were treated with PBS or 50 μM biotin. After 30 min of incubation, cells were harvested and lysed. Biotinylated proteins were extracted by streptavidin affinity purification. The input lysate (lysate), the remnant lysate (remnant) after affinity purification and the pulldown samples (elution) were analyzed by (A) SDS-PAGE and (B) western blot. The Q-TOF analysis of the peptides resulting from tryptic digestion of biotinylated proteins revealed numerous potential interactors of SETD3 listed in Table 2.

As shown in Fig. 4B, minimal endogenous biotinylated proteins were isolated from control cells transfected with an empty plasmid (File S3). Biotin addition to the culture medium resulted in non-specific overlabeling of endogenous proteins, potentially overloading the streptavidin beads (Fig. 4B). Notably, efficient protein enrichment occurred even without exogenous biotin, facilitating the identification of biotinylated proteins in that cell lysate. In total, LC-MS/MS identified 23 unique human proteins in the streptavidin-purified samples (Table 2, File S3), including SETD3 and actin. Interestingly, compared to cells incubated with biotin, a more diverse set of proteins were identified from the sample using biotin-free cell lysates. This difference likely arose from: (i) excessive protein labeling with added biotin, (ii) overloading of beads with a few highly labeled proteins, and (iii) insufficient protein digestion due to excessive biotinylation of lysine residues (Li et al., 2021).

Table 2 The unique proteins identified in the streptavidin-purified samples of HAP1 cell lysates from the TurboID proximity labeling experiments.

Identified Proteins	
Accession	Acronym	Description	mW (Da)	
+BIOTIN	
NP_115609.2	SETD3	actin histidine N methyltransferase isoform	67,215	
NP_001243650.1	CCT4	T complex protein 1 subunit delta isoform b	54,685	
NP_006421.2	CCT4	T complex protein 1 subunit delta isoform a	57,887	
NP_006187.2	PCBP1	poly rC binding protein 1	37,473	
NP_001258978.1	TRAP1	heat shock protein 75 kDa mitochondrial	74,220	
+PBS	
NP_115609.2	SETD3	actin histidine N methyltransferase	67,215	
NP_001407.1	EIF4A1	eukaryotic initiation factor 4A I isoform 1	46,124	
NP_001243650.1	CCT4	T complex protein 1 subunit delta isoform b	54,685	
NP_004631.1	DDX39B	spliceosome RNA helicase DDX39B	48,960	
XP_011539528.1	PHGDH	D 3 phosphoglycerate dehydrogenase isoform X1	64,653	
NP_110379.2	TCP1	T complex protein 1 subunit alpha isoform a	60,305	
NP_001258898.1	HSP90AB1	heat shock protein HSP 90 beta isoform a	83,212	
NP_001092090.1	PCBP2	poly rC binding protein 2 isoform c	34,894	
NP_001185771.1	CCT2	T complex protein 1 subunit beta isoform 2	52,684	
NP_001243064.1	CSE1L	exportin 2 isoform 2	103,813	
NP_001186883.1	ACTG1	actin cytoplasmic 2	41,765	
NP_001001.2	RPS6	40S ribosomal protein S6	28,663	
XP_005261192.1	PFKL	ATP dependent 6 phosphofructokinase l	87,620	
NP_001001937.1	ATP5F1A	ATP synthase subunit alpha mitochondrial	59,713	
NP_001008800.1	CCT3	T complex protein 1 subunit gamma isoform c	56,395	
NP_001231867.1	TXN	thioredoxin isoform 2	9,445	
NP_005727.1	ACTR1A	alpha centractin	42,586	
NP_004095.4	FASN	fatty acid synthase	273,251	
NP_001013757.1	FOXB2	forkhead box protein B2	45,551	
NP_001316603.1	CAND1	cullin associated NEDD8 dissociated protein 1 isoform 2	133,529	
NP_001003699.1	RREB1	ras responsive element binding protein 1 isoform 1	187,432	
Notes.

HEK293T cells that overexpressed the SETD3 enzyme, fused at either its C- or N-terminus with the TurboID biotin ligase, were treated with PBS or 50 μM biotin. Following a 30-minute incubation period, the cells were harvested and lysed. Biotinylated proteins were then isolated by streptavidin affinity purification, subjected to tryptic digestion, and the resulting peptides were identified by Q-TOF mass spectrometry.

Similar experiments were conducted in HeLa and HAP1 cells lacking SETD3. However, protein biotinylation was significantly lower in these cell lines compared to HEK293T cells, possibly due to reduced expression of the recombinant SETD3-TurboID fusion proteins.

Several proteins may be the substrates of SETD3

The SET domain, a common feature of protein methyltransferases that modify lysine residues, often exhibits high substrate specificity due to recognition of flanking sequences surrounding the targeted lysine (Dai et al., 2019). Similarly, SETD3 appears to exhibit this characteristic.

Previous studies demonstrated that the β-actin peptide (residues 66–88) fits into a specific groove on SETD3, with residues Ile71, His73, and Trp79 occupying three hydrophobic pockets (Guo et al., 2019; Dai et al., 2019). Substitution of Ile71 and Trp79 in the β-actin peptide with shorter or charged residues significantly reduced its suitability as a substrate (Bilgin et al., 2022; Al-Fakhar et al., 2023). These findings suggest that the mechanisms governing β-actin peptide binding likely apply to other SETD3 substrates as well.

Therefore, to identify the most promising potential substrates from the proteins identified in fluorography and TurboID experiments, we employed the following criteria: (i) the candidate protein should be [2H]-methylated at a histidine residue, (ii) the candidate protein’s molecular weight should correspond to one of the bands visualized in the fluorography experiment, and (iii) the sequence flanking the methylated histidine should contain relatively hydrophobic residues at positions −2 and +5 relative to the histidine, analogous to Ile71 and Trp79 in the β-actin peptide. Based on these criteria, fourteen different proteins, including β-actin, were selected as potential SETD3 substrates, along with their sequences containing the putative methylation site (Table 3). Notably, some of these potential substrates, such as actin and fatty acid synthase, were identified in both fluorography and TurboID experiments.

Table 3 Potential protein substrates of SETD3 selected among proteins identified in MS/MS experiments.

Accession	Acronym	Description	mW (Da)	Peptide sequence	
NP_001092.1	ACTB	actin, cytoplasmic 1	41,709	KYPIEHGIVTNW	
NP_005727.1	ACTR1A	alpha centractin	42,586	RYPMEHGIVKDW	
NP_001531.1	HSPB1	heat shock protein beta 1	22,768	LDVNHFAPDEL	
NP_004095.4	FASN	fatty acid synthase	273,251	EPTAHAFVSTL	
NP_001361230.1	IARS1	isoleucine tRNA ligase cytoplasmic	141,287	IVVIHQDPEAL	
NP_001376654.1	GLOD4	glyoxalase domain containing protein 4	39,973	GGVDHAAAFGRI	
NP_001352825.1	NACA	nascent polypeptide associated complex subunit alpha	205,293	DGISHTSALAP	
NP_001348.2	DHX9	ATP dependent RNA helicase A	140,868	RFSDHVALLSV	
NP_001072989.1	KTN1	kinectin isoform a	156,178	EPLIHATTYIP	
NP_036565.2	SF3B1	splicing factor 3B subunit 1	145,737	VEIIEHGLVDE	
NP_005436.1	SMC3	structural maintenance of chromosomes protein 3	141,453	LALLHEGTGPR	
NP_005984.3	TBCD	tubulin specific chaperone D	132,515	RPVTDHLDEQAV	
NP_001290176.1	SKIC8	WD repeat containing protein 61	33,559	KIYDVQHANLAG	
NP_006436.3	PRPF8	pre mRNA processing splicing factor 8	273,424	PVLYHITGAIS	
Notes.

The histidine residues of the listed proteins that were found to be [2H]-methylated in MS/MS experiments are shown in bold.

Alpha-centractin is a new substrate of SETD3

To verify the activity of human SETD3 towards its potential substrates in vitro, peptides were synthesized and tested as acceptors of [3H]methyl from [3H]SAM in a radiochemical assay. As shown in Fig. 5, SETD3 catalyzed the methylation of β-actin and α-centractin peptides. Interestingly, the enzyme also displayed activity towards a peptide derived from the isoleucine tRNA ligase, albeit at a significantly lower level.

Figure 5 SETD3-mediated methylation of synthetic peptides containing histidine residues.

Recombinant human SETD3 protein (20 μg) was incubated with [3H]SAM (approximately 300–450 × 103 cpm) and 1 mM (100 nmol) of synthetic peptides (sequences shown in Table 3) for 4 h at 37 °C. Methylated peptides were separated from [3H]SAM using Dowex 50W ion-exchange columns. The radioactivity (cpm) shown has been normalized by subtracting background levels. Values are the means ± SD (error bars) of three independent experiments. When no error bar is shown, the error is smaller than the width of the line. ACTB, actin, cytoplasmic 1; ACTR1A, α-centractin; HSPB1, heat shock protein β 1; FASN, fatty acid synthase; IARS1, isoleucine tRNA ligase cytoplasmic; GLOD4, glyoxalase domain containing protein 4; NACA, nascent polypeptide associated complex subunit α; DHX9, ATP dependent RNA helicase A; KTN1, kinectin isoform a; SF3B1, splicing factor 3B subunit 1; SMC3, structural maintenance of chromosomes protein 3; TBCD, tubulin specific chaperone D; SKIC8, WD repeat containing protein 61; PRPF8, pre mRNA processing splicing factor 8.

To further confirm the specific methylation of α-centractin, a radiochemical assay was performed using recombinant human α-centractin as the substrate, alongside β-actin as a positive control. The α-centractin protein was overexpressed and purified from Escherichia coli using the same established protocol previously employed for β-actin production. As shown in Fig. 6, wild-type SETD3 methylated recombinant α-centractin, demonstrating its suitability as a substrate in vitro. Notably, methylation was absent in the presence of R253A SETD3, a catalytically inactive mutant, confirming the enzyme’s specific activity. However, the methylation efficiency of wild-type SETD3 towards α-centractin was considerably lower compared to β-actin, suggesting that the latter serves as a significantly better substrate for the enzyme. This can be attributed to the kinetic properties of SETD3 in the presence of α-centractin (Table 4). The enzyme displayed a considerably lower affinity for α-centractin (3.827 µM) compared to β-actin (0.752 µM) (Kwiatkowski et al., 2018). Furthermore, its kcat with α-centractin and a saturating concentration of SAM (0.31 s−1) was roughly half that estimated for β-actin (0.65 s−1). The enzyme’s affinity for SAM was also somewhat reduced with α-centractin (0.242 µM) compared to β-actin (0.116 µM) (Kwiatkowski et al., 2018). However, it is important to note that we conducted these kinetic studies using only sub-saturating concentrations of α-centractin, as the substrate tended to precipitate at higher concentrations (Fig. S5).

Figure 6 In vitro methylation of β-actin and α-centractin by wild-type and R253A variant of SETD3 protein.

Wild-type human SETD3 and its enzymatically inactive variant R253A were overexpressed in E. coli and purified by affinity chromatography using nickel-Sepharose (HisTrap FF Crude), as detailed in the ‘Materials and Methods’ section. Recombinant SETD3 protein (0.5 or 1.5 μg) or its R253A mutant (1.5 μg) were incubated with 1 μM (100 pmol, ≈ 290 × 103 cpm) [1H + 3H]SAM and 2 μM (200 pmol) of purified human β-actin or α-centractin at 37 °C for the indicated times. Proteins were then precipitated with 10% trichloroacetic acid (TCA) to measure the incorporation of radioactivity. Values are the means ± SD (error bars) of three independent experiments. When no error bar is shown, the error is smaller than the width of the line.

To confirm the specific methylation of H77 in recombinant human α-centractin, we incubated the protein with either homogeneous recombinant human SETD3 or R253A mutant enzymes. This incubation was performed in the presence of [2H] SAM. Following the reaction, the proteins in the assay mixture were tryptic-digested. The resulting α-centractin peptides were then analyzed using nano-UPLC-Q-TOF. Each MS analysis achieved at least 37% sequence coverage of α-centractin. The peptide YPMEHGIVK (M + H = 1,090.5804 Da), which contains the H77 residue, was the only trideuterium-methylated peptide detected by the PLGS 2.4 software (File S4). Manual inspection of the MS/MS spectra of both the parent ion and its fragments definitively confirmed the presence of trideuterium-methylated H77 (Fig. 7). No trideuterium-methylation of H77 was detected in the YPMEHGIVK peptide (M + H = 1,073.5479 Da) derived from α-centractin protein that had been incubated with the R253A mutant protein (see Fig. 7).

Table 4 Kinetic properties of human SETD3 protein.

Substrate		Vmax nmol min1 mg1	KMμM	kcat min −1	
α-centractin		4.328 ± 0.399	3.827 ± 0.637	0.31	
S-adenosyl-L-methionine		1.509 ± 0.088	0.242 ± 0.046	0.11	
Notes.

Kinetic properties were determined with the use of purified recombinant N-terminal His6-tagged SETD3 protein. Determinations for S-adenosyl-L-methionine (SAM) were performed with SETD3 preparations (0.5 μg protein, 140 nM) that were incubated for 5 min at 37 °C in the reaction mixture containing 4 μM recombinant α-centractin and variable concentrations of [1H + 3H]SAM (0.1–2.0 μM, 330,000 cpm), while the measurements for α-centractin (0.2–5 μM) were obtained following a 5-min incubation of SETD3 in the presence of 1 μM concentration of [1H + 3H]SAM (50 pmol, 330,000 cpm). In all experiments, the reaction mixture contained the homogenous recombinant SAH nucleosidase (0.8 μg protein, 300 nM, E. coli) and adenine deaminase (2 μg protein, 300 nM, B. subtilis) to prevent S-adenosyl-L-homocysteine (SAH) accumulation. Values are the means of three separate experiments. The values for standard deviation (SD) are also given.

Figure 7 Deconvoluted Q-TOF spectra revealing H77 as the methylation site in α-centractin peptides.

The spectra compare non-modified (+ R253A) and deuterium-methylated α-centractin peptides (+ SETD3) produced by either active human SETD3 protein or its inactive R253A variant. For the reaction, 20 μg of recombinant human α-centractin was incubated for 120 min at 37 °C with 30 μM [2H]SAM, 1.5 μg SAH nucleosidase, and either 1.5 μg SETD3 or 1.5 μg R253A protein. Following the reaction, α-centractin was trypsin-digested and analyzed by tandem mass spectrometry as described in the ‘Materials and Methods’ section. Trideuterium-methylated peptides were detected using ProteinLynx Global Server 2.4 software (Waters, USA) by examining a mass shift of +17.03448 Da corresponding to trideuterium methylation of Cys, Asp, Asn, His, Lys, Arg, Glu, and Gln residues. Each MS/MS analysis achieved at least 37% α-centractin sequence coverage. The H77-containing peptide, YPMEHGIVK (M + H = 1073.5448 Da, y9 ion), was the sole trideuterium-methylated peptide (M + H = 1090.5793 Da, y9 ion) observed in the presence of SETD3, while it remained unmethylated in the R253A control reaction. A mass shift of +17.03448 Da was exclusively detected in His-containing peptide fragments (y5, y6, y7, y8, y9), indicating that the H77 residue is the specific site of methylation. Results from a representative experiment are shown; two independent labeling reactions were performed using [1H]SAM or [2H]SAM.

Discussion

While β-actin is a known substrate for SETD3 in human cells, there may be others

Previous research, including work from our laboratory, identified SETD3 as an enzyme that specifically methylates H73 residue on β-actin (Kwiatkowski et al., 2018; Wilkinson et al., 2019). Currently, β-actin remains the only confirmed substrate for this methyltransferase. However, SETD3 has been implicated in various cellular processes apparently unrelated to actin regulation, such as myocyte differentiation, cell cycle control, and the response to hypoxic conditions (Eom et al., 2011; Kim et al., 2011; Cohn et al., 2016). Additionally, studies have shown SETD3 localization within the nucleus of mammalian cells (Eom et al., 2011; Cohn et al., 2016), hinting at the potential for different substrates in various cellular compartments. This raises a key question: is β-actin the sole substrate for SETD3 methyltransferase activity?

Here, we show that SETD3 methylates several polypeptides in human HAP1 cells that are distinct from β-actin. These novel substrates appear to be present in minute quantities. This is evidenced by their very weak labeling and the faint protein bands observed during SDS-PAGE analysis in the gel regions where they were visualized (see Fig. 2 and Fig. S4). Furthermore, it is highly unlikely that these minor SETD3 substrates are, in fact, major substrates for other histidine-specific methyltransferases. Unlike lysine- or arginine-specific methyltransferases, no cross-reactivity has been observed among histidine-specific methyltransferases to date. Every known enzyme in this class—including SETD3, METTL9, METTL18, and C9orf41—catalyzes the methylation of its own specific substrates. Due to the low abundance of these novel substrates compared to β-actin, their detection was only possible by using SETD3-deficient cells as a substrate source for in vitro labeling experiments dependent on SETD3 activity. This might explain why these alternative substrates have not been observed previously in experiments using lysates from cells expressing endogenous SETD3 (Wilkinson et al., 2019).

Mass spectrometry analysis identified numerous proteins within the labeled bands, but only a few exhibited histidine residue methylation. In vitro methylation experiments using synthetic peptides derived from candidate proteins (see Fig. 5) revealed detectable methylation of the Isoleucyl-tRNA synthetase 1 (IARS1) peptide. This suggests that IARS1 might be a SETD3 substrate present in the labeled bands. However, attempts to produce the recombinant IARS1 protein were unsuccessful, which prevented us from further testing it as a substrate for SETD3 in vitro. Also noteworthy is that the absence of SETD3-dependent methylation of the remaining tested peptides does not definitively exclude the possibility that their parent proteins are still substrates of the methyltransferase. Synthetic peptides have already been shown to be poor substrates for some histidine methyltransferase which may efficiently methylate full-length proteins in vitro or in vivo (Webb et al., 2010; Davydova et al., 2021; Małecki et al., 2021). This likely occurs because some methyltransferases require complex interactions with the entire protein substrate for proper recognition and efficient methylation.

ACTR1A is a new substrate for SETD3 in vitro

We found that all SETD3 substrates visualized in fluorography were involved in the formation of protein complexes that also included the SETD3 enzyme (see Figs. 3B–3D and File S2). This aligns with previous studies demonstrating that: (i) endogenous SETD3 in skeletal muscle is tightly bound to myofibrils, forming a relatively stable complex with myofibrillary proteins (Vijayasarathy & Rao, 1987; Kwiatkowski et al., 2018), and (ii) SETD3 interacts with a large number (around 170) of different intracellular proteins in mammalian cells (Cohn et al., 2016). The exact mechanisms and reasons behind the formation of these complexes remain unclear. However, it is tempting to speculate that these interactions might play a role in how SETD3 recognizes substrates and performs its enzymatic activity.

Our work has established that α-centractin (ACTR1A) physically interacts with SETD3 and serves as a substrate for its methyltransferase activity. This conclusion is supported by two findings: (i) proximity labeling experiments using HEK293T cell lysates identified ACTR1A as a protein interacting with SETD3 (see Fig. 4) and (ii) both human recombinant α-centractin and a synthetic peptide derived from this protein were methylated in enzymatic assays (Figs. 5–7). The observation that α-centractin is a SETD3 substrate is unsurprising, considering its close resemblance to β-actin in both amino acid sequence and spatial structure (Fig. 8). However, more importantly, our data demonstrates that SETD3 methylates β-actin significantly more efficiently than ACTR1A. This aligns with the current understanding of SETD3 substrate specificity, which likely relies on precise recognition of the amino acid sequence surrounding the methylation site (Guo et al., 2019; Dai et al., 2019).

Figure 8 Structure and sequence similarity between human β-actin and α-centractin.

(A) Amino acid sequence alignment of human β-actin (Actin, cytoplasmic 1, NP_001092.1) and α-centractin (NP_005727.1). The sequences were obtained from the NCBI Protein database, and the percentage of amino acid identities between them is equal to 53% (BLASTP 2.16.0+). The level of conservation of residues is indicated by the sequence logo. The sequences corresponding to synthetic peptides used for testing the activity of SETD3 (see Table 3) are underlined in the sequence logo. Ribbon representations of (B) human β-actin (PDB: 2BTF), (C) human α-centractin (AlphaFold: AF-P61163-F1) and (D) the superposition of both protein structures, highlighting a clearly similar fold architecture (a root mean square deviation of 2.709 Å). (B) The sequence corresponding to the synthetic peptide ACTB (see Table 3) is shown (lime ribbon), including the targeted histidine (red sticks), while (C) the sequence corresponding to the peptide ACTR1A is colored in magenta and the histidine is shown in cyan. All models were prepared using UCSF ChimeraX (Meng et al., 2023).

Potential physiological role of ACTR1A methylation

α-Centractin (or actin-related protein Arp 1) is a crucial subunit of dynactin. Dynactin is a key component of the cytoplasmic dynein motor complex, which plays a central role in transporting various cellular cargo along microtubules in eukaryotic cells (Urnavicius et al., 2015; Reck-Peterson et al., 2018). Notably, among all actin-related proteins in eukaryotes, α-centractin exhibits the highest similarity to conventional β-actin. It shares the ability to bind and hydrolyze ATP, and can spontaneously polymerize into short filaments resembling, but not identical to, actin filaments (Bingham & Schroer, 1999). Importantly, an α-centractin filament composed of eight monomers can bind a single β-actin monomer, forming the central core of the 23-subunit (11 distinct proteins) dynactin complex.

Previous research has shown that methylation of a specific histidine residue (H73) in β-actin promotes its polymerization. We hypothesize that methylation of the corresponding histidine residue (H77) in α-centractin by SETD3 might similarly stimulate filament formation, thereby facilitating the assembly of the dynactin complex. Consequently, a loss of SETD3 activity could potentially impact not only actin polymerization but also dynein-dependent processes like intracellular transport.

Conclusions

In conclusion, our studies reveal that β-actin, while a major substrate, is not the only target for SETD3 methyltransferase activity in human cells. The enzyme can methylate several additional intracellular polypeptides distinct from β-actin. Furthermore, we provide evidence supporting previous findings that SETD3 forms stable complexes with intracellular proteins, potentially including its substrates. Significantly, α-centractin emerges as one such interactor, which does not only interact with SETD3 but is also methylated by the enzyme. This suggests that SETD3 activity may play an important role not only in actin cytoskeleton assembly but also in dynein-dependent intracellular transport processes.

Supplemental Information

Supplemental Information 1 SDS-PAGE analysis of purified recombinant human SETD3 (A) and R253A variant (B)

Recombinant human SETD3 protein (A) and its enzymatically inactive variant R253A (B) were purified by affinity chromatography using nickel-Sepharose (HisTrap FF crude), as detailed in the ‘Materials and Methods’ section. For SDS-PAGE analysis, 11 μ L of each eluted fraction was loaded onto a 12% polyacrylamide gel, electrophoresed, and the gel was stained with colloidal Coomassie Brilliant Blue. Arrows indicate protein bands corresponding to the purified SETD3 or R253A protein. M, prestained protein marker; L, cell-free lysate of E. coli loaded on the column; FT, flow through; W, wash; Fractions 60 to 300 were eluted with the indicated concentrations of imidazole.

Supplemental Information 2 SDS-PAGE analysis of purified recombinant human α-centractin

Recombinant human α-centractin (ACTR1A) was purified to homogeneity by affinity chromatography using nickel-Sepharose (HisTrap FF crude), as described in the ‘Materials and Methods’ section. For SDS-PAGE analysis, 15 μ L of each eluted fraction was loaded onto a 12% polyacrylamide gel. The proteins were separated by electrophoresis, and the gel was stained with colloidal Coomassie Brilliant Blue. The arrow indicates the protein band corresponding to the purified α-centractin. M, prestained protein marker; L, urea-washed inclusion bodies of E. coli loaded on the column; FT, flow through; W, wash; Fractions 60 and 300 were eluted with the indicated concentrations of imidazole.

Supplemental Information 3 Lack of SETD3 protein expression in SETD3-deficient HEK 293T and HeLa cell lines compared to control cells

(A) SETD3 expression in SETD3-deficient HEK 293T cells. Western blot analysis was performed to compare SETD3 protein levels in eight clonal HEK 293T cell lines deficient in SETD3 with control cell lines: HEK293T wild-type (WT) cells (A1 and A2) and the HAP1 cell line (WT-C6). (B) SETD3 expression in SETD3-deficient HeLa cells. Western blot analysis was performed to compare SETD3 protein levels in seven clonal HeLa cell lines deficient in SETD3 with control cell lines: HeLa wild-type cells (WT-21) and the HAP1 cell line (WT-C6). The analyses were performed using 100 μ g of the cell lysate protein loaded per lane. Rabbit primary antibody against the human SETD3 (ab174662, Abcam) along with a horseradish-peroxidase-conjugated goat anti-rabbit secondary antibody were used for detection. Enhanced chemiluminescence (ECL) was used for visualization. Note: A non-specific band observed at approximately 36 kDa is consistent with the manufacturer’s specifications for the primary antibody.

Supplemental Information 4 SETD3 methylates proteins different from β-actin in human cells

Cell-free lysates of human wild-type (WT) or SETD3 knockout (KO) HAP1 cells (230 μ g protein) were preincubated with [1 H]SAM for 30 min and then gel-filtered to remove [1H]SAM. Next, [3H]SAM (1x106 cpm) was added to the reaction mixture and the radiolabeling of the protein was started by adding homogeneous recombinant SETD3 or its inactive R253A variant (10 μ g protein). The reaction mixtures were incubated for 60 or 180 min at 37 °C and resolved by SDS-PAGE (A, C). The [3H] methylated proteins were detected by fluorography for 4 weeks at −80 °C as described in Materials and Methods (B, D). Either 30 μ g (A, B) or 60 μ g (C, D) protein were loaded on the gels. The arrows show the methylated polypeptides that differentiate the lysates incubated with inactive (R253A) and active (SETD3) methyltransferase. M, prestained protein marker.

Supplemental Information 5 Michaelis-Menten equation curves for reactions catalyzed by human SETD3

Michaelis-Menten plots for (A) α-centractin and (B) S-adenosyl-L-methionine (SAM) are shown. The activity of methyltransferase was followed radiochemically by measuring the incorporation of the [3 H]methyl group into α-centractin protein. Determinations for SAM were performed with SETD3 preparations (0.5 μ g protein, 140 nM) that were incubated for 5 min at 37 °C in the reaction mixture containing 4 μ M recombinant α-centractin and variable concentrations of [1H+3H] SAM (0.1–2.0 μ M, 330,000 cpm), while the measurements for α-centractin (0.2–5 μ M) were obtained following a 5-min incubation of SETD3 in the presence of 1 μ M concentration of [1H+3H] SAM (50 pmol, 330,000 cpm). In all experiments, the reaction mixture contained the homogenous recombinant SAH nucleosidase (0.8 μ g protein, 300 nM, E. coli) and adenine deaminase (2 μ g protein, 300 nM, B. subtilis) to prevent S-adenosyl-L-homocysteine (SAH) accumulation. Values are the means of three separate experiments. Values are the means ± SD (error bars) of three independent experiments. The curves were plotted employing Origin 2025 software (OriginLab, USA) and nonlinear regression analysis.

Supplemental Information 6 Proteins identified in gel bands submitted to trypsin digestion and MS/MS analysis

The proteins methylated (deuterated) at histidine residues are shown on a yellow background. All identified peptides of the modified proteins are also listed.

OK column indicates whether the protein or peptide match has been determined as good (2), possible (1) or poor (0).

Supplemental Information 7 Proteins identified in 10-fold concentrated fraction 15 (Superdex 200) that was digested with trypsin and submitted to MS/MS analysis

The proteins methylated (deuterated) at any residues are shown on a yellow background, whereas those methylated at the histidine residue are shown on a blue background. All identified peptides of the modified proteins are also listed. OK column indicates whether the protein or peptide match has been determined as good (2), possible (1) or poor (0).

Supplemental Information 8 Proteins identified in an eluate of streptavidin magnetic beads incubated with lysates of HEK293T cells

OK column indicates whether the protein or peptide match has been determined as good (2), possible (1) or poor (0).

Supplemental Information 9 Q-TOF analysis of non-modified and methylated alpha-centractin peptides in the presence of wild-type and R253A SETD3

We are very grateful to colleagues from the Department of Molecular Biology (University of Warsaw) for making the FPLC apparatus available for protein purification. The manuscript was proofread with the aid of an artificial intelligence tool, Google Gemini 2.5.

Additional Information and Declarations

Competing Interests

Author Contributions

Data Availability

The authors declare there are no competing interests.

Apolonia Witecka conceived and designed the experiments, performed the experiments, analyzed the data, prepared figures and/or tables, authored or reviewed drafts of the article, and approved the final draft.

Paulina Emmel conceived and designed the experiments, performed the experiments, analyzed the data, authored or reviewed drafts of the article, and approved the final draft.

Klaudia Ślusarczyk performed the experiments, analyzed the data, authored or reviewed drafts of the article, and approved the final draft.

Julia Z. Kamińska performed the experiments, analyzed the data, authored or reviewed drafts of the article, and approved the final draft.

Michał  Zaród conceived and designed the experiments, performed the experiments, analyzed the data, authored or reviewed drafts of the article, and approved the final draft.

Takao Ishikawa conceived and designed the experiments, performed the experiments, analyzed the data, authored or reviewed drafts of the article, and approved the final draft.

Jakub Drożak conceived and designed the experiments, performed the experiments, analyzed the data, prepared figures and/or tables, authored or reviewed drafts of the article, and approved the final draft.

The following information was supplied regarding data availability:

The raw results of LC-MS/MS analyses are available in the Supplementary Files.

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
