# Peer review of "Alpha-centractin is a novel substrate of SETD3 methyltransferase in vitro"

_PeerJ, doi:10.7717/peerj.20195_

## Round 0.1 · original submission · Major Revisions

Thank you for submitting this work to PeerJ. The reviewers comments are enclosed. As you will see, reviewer-3 is seeking additional experimental work, an opinion which I fully endorse.

These extra experiments are deemed to be essential to validate your work by the inclusion of additional controls, and here I refer specifically to point 1. Figure 2 is the crux of the paper and it needs much more development to convince.

Some comparison with other histidine methyltransferases is warranted here. The data in figure 2 clearly suggest that the other bands are very minor substrates compared to STED3. This may simply reflect that they are lesser efficient substrates, and that the primary methyltransferase acting upon them is a different family member. At the very last, this issue must be discussed.

I am concerned about the TURBO data shown in panel B. Why do the profiles of lysate, remnant and eluate all look so similar? The STRING analysis is not a validation; while it is interesting it does not prove association. I would like to see some kind of co-IP approach here.

Figure 4 is not necessary. Please confirm that the point mutants referred to as generated here have been fully sequenced.

If you require more time than the standard 25 days suggested below, please advise the editorial team. I realise this may be necessary as more work is needed.

Reviewer 1 ·

Basic reporting

The authors found a new substrate for mammalian SETD3, the finding would be of general interest to the protein PTM field

Experimental design

The authors not only identify the cognate substrate for SETD3, but also designed the mutations to identify the SETD3 residues that are involved in substrate recongtion and/or catalysis, which
well supports the model. The figures are clear and informative.

Validity of the findings

The paper is well written and succinct. I support publication.

Reviewer 2 ·

Basic reporting

no comment

Experimental design

The experimental design and protocols have been described well and allow other researchers to reproduce the experimental work.

Validity of the findings

Alpha-centractin is identified as a new substrate for histidine methyltransferase SETD3 using a complemental set of biochemical and cellular assays. It is stated that experiments shown in Figures 6 and 7 were run as single experiments with similar results obtained in two independent experiments. It is not clear why standard deviations are not shown for the two independent experiments.

Additional comments

Some minor changes are required.

The structures of His/meHis in Fig. 1A need to have the L-stereochemistry specified.

Line 88, Npi-specific should be replaced by Ntele-specific

While time-course experiments shown in Fig. 7 are valuable, the work would benefit from enzyme kinetics assays that provide more solid comparison between beta-actin and alpha-centractin substrates of SETD3.

·

Basic reporting

The manuscript by Witecka et al focuses on the identification of novel substrates of SETD3, a histidine MTase originally found to methylate a single His residue in actin. Basically, the paper consists of two parts. Firstly, it is investigated which proteins are methylated when an extract from SETD3 KO cells is incubated with recombinant SETD3 in the presence of radiolabelled SAM. These efforts led to the identification of several labelled bands (in a protein gel) corresponding to possible SETD3 substrates, but, unfortunately, attempts to identify these substrates were not successful. Secondly, the TURBO ID method is used to identify putative SETD3 interactants, and peptides derived from these interactants were tested as SETD3 substrate, leading to the identification of alpha-centractin as an in vitro substrate of SETD3. The manuscript is well written and easy to understand, and shows a good knowledge of the scientific literature. The experimental data are well presented, and adequately discussed.
Specific comments:
1. I am not really convinced that the labelled bands in Fig. 2B really represent SETD3 substrates. There are some very straightforward experiments the authors should do to assess this further. First, instead of using BSA as a negative control, the authors should use a catalytically inactive mutant of SETD3. This will hopefully exclude the possibility that the labelled bands represent contaminating proteins from the SETD3 prep. Second, the authors should compare extracts from wild-type and SETD3 KO cells. If the bands represent true SETD3 substrates that show a high degree of methylation occupancy in WT cells, one would expect that the bands observed with KO cells are stronger than for WT cells (as there is more unmethylated substrate present).
2. Based on the TURBO-ID results, alpha-centractin is identified as an in vitro substrate of SETD3. However, this result is not very groundbreaking, as a simple BLAST search using as query the SETD3-targetted sequence in actin, reveals alpha-centractin and beta-centractin as the best candidate substrates for SETD3 (besides so-called POTE proteins, which show a rather limited expression in most cells). It is therefore also surprising that beta-centractin was not tested as a SETD3 substrate.
3. This manuscript would have been substantially strengthened by a demonstration that alpha-(and/or beta-) centractin is subject to SETD3-mediated methylation in cells.
4. First page, l. 75. References to papers showing that human METTL18 targets RPL3 are lacking (the supplied references are to yeast Hpm1).

Experimental design

Satisfactory

Validity of the findings

My major issue with this manuscript is that a large fraction of the experiments are inconclusive and that that it basically only contains a single result, namely that SETD3 is capable of methylating alpha-centractin in vitro.

Additional comments

None

---

## Round 0.2 · accepted · Accept

The reviewers are happy with your revised manuscript, and I am pleased to recommend acceptance.

Reviewer 2 ·

Basic reporting

no comment

Experimental design

no comment

Validity of the findings

no comment

Additional comments

The authors have been responsive to my earlier suggestions. I have no additional requests.

·

Basic reporting

Adequate

Experimental design

Adequate

Validity of the findings

Adequate

Additional comments

The authors have undertaken substantial experimental efforts to address my comments. Although not all of these were successful, the manuscript has been substantially improved, and I am happy to recommend it for publication.

Note: I was unable to locate the figure legends for the suppl. figures.